

# Lymph node detection in MR Lymphography: false positive reduction using multi-view convolutional neural networks

Oscar A. Debats, Geert J.S. Litjens and Henkjan J. Huisman

Department of Radiology and Nuclear Medicine, Radboudumc, Nijmegen, The Netherlands

## ABSTRACT

**Purpose**. To investigate whether multi-view convolutional neural networks can improve a fully automated lymph node detection system for pelvic MR Lymphography (MRL) images of patients with prostate cancer.

**Methods**. A fully automated computer-aided detection (CAD) system had been previously developed to detect lymph nodes in MRL studies. The CAD system was extended with three types of 2D multi-view convolutional neural networks (CNN) aiming to reduce false positives (FP). A 2D multi-view CNN is an efficient approximation of a 3D CNN, and three types were evaluated: a 1-view, 3-view, and 9-view 2D CNN. The three deep learning CNN architectures were trained and configured on retrospective data of 240 prostate cancer patients that received MRL images as the standard of care between January 2008 and April 2010. The MRL used ferumoxtran-10 as a contrast agent and comprised at least two imaging sequences: a 3D T1-weighted and a 3D T2$^\star$-weighted sequence. A total of 5089 lymph nodes were annotated by two expert readers, reading in consensus. A first experiment compared the performance with and without CNNs and a second experiment compared the individual contribution of the 1-view, 3-view, or 9-view architecture to the performance. The performances were visually compared using free-receiver operating characteristic (FROC) analysis and statistically compared using partial area under the FROC curve analysis. Training and analysis were performed using bootstrapped FROC and 5-fold cross-validation.

**Results**. Adding multi-view CNNs significantly ($p < 0.01$) reduced false positive detections. The 3-view and 9-view CNN outperformed ($p < 0.01$) the 1-view CNN, reducing FP from 20.6 to 7.8/image at 80% sensitivity.

**Conclusion**. Multi-view convolutional neural networks significantly reduce false positives in a lymph node detection system for MRL images, and three orthogonal views are sufficient. At the achieved level of performance, CAD for MRL may help speed up finding lymph nodes and assessing them for potential metastatic involvement.

Corresponding author
Oscar A. Debats, debats@gmail.com

## INTRODUCTION

Approximately 220,000 men are diagnosed with prostate cancer (PCa) in the USA each year, and 27,000 die as a consequence of the disease (*Siegel, Miller & Jemal, 2017*). Assessment of lymph node status is crucial in determining the best treatment for a patient. MR Lymphography (MRL) is currently the only imaging modality with a reported negative predictive value over 95% for detection of metastatic lymph nodes (*Fortuin et al., 2014*). In a prospective study using histopathology as a reference standard, MRL was shown to be an accurate imaging technique with sensitivities up to 91%, at 98% specificity (*Harisinghani et al., 2003*). In contrast, conventional MRI has a substantially lower diagnostic accuracy, according to a meta-analysis reporting a pooled sensitivity of 39% with a pooled specificity of 82% (*Hövels et al., 2008*).

MRL is MR imaging with a contrast agent based on ultra-small super-paramagnetic particles of iron oxide (USPIO), which result in signal intensity differences between metastatic and normal lymph node tissue (*Heesakkers et al., 2008*; *Daldrup-Link, 2017*). Analysis of MRL images by radiologists is very time-consuming, with average reading times up to 80 min, and requires readers with a very high level of experience for the assessment of the images (*Thoeny et al., 2009*).

MRL interpretation time, as well as its dependence on high experience level, can be reduced via computer-aided detection (CAD) systems. The first goal of CAD is to reduce search time by detecting all lymph nodes visible in the MRL images, whether healthy or diseased, and present each one subsequently to the human reader for interpretation. In other words, such a CAD system is not intended to detect disease, but rather to detect a certain type of anatomical structure (i.e., the lymph nodes). A modest number of studies have been published on the development of lymph node CAD systems, most of them using CT imaging rather than MR imaging (*Kitasaka et al., 2007*; *Roth et al., 2014*; *Roth et al., 2016*; *Seff et al., 2014*).

We have previously shown that a feature-based CAD system levering anatomical models with an atlas approach can, after being properly trained, sensitively detect lymph nodes in abdominal MR imaging (*Meijs, Debats & Huisman, 2015*; *Debats et al., 2016*). However, the number of false positive detections was too high for clinical application. Various types of conventional image features were explored but we could not find a set that was able to discriminate true lymph nodes from false positive detections with high enough accuracy.

Recently, deep learning and, more specifically, convolutional neural networks (CNNs) have been shown to outperform conventional machine learning strategies based on hand-crafted features. Deep learning is successfully applied in the field of medical image analysis outperforming classical machine learning and achieving expert performance (*De Fauw et al., 2018*; *Litjens et al., 2017*). Examples of the application of CNNs in abdominal imaging: kidney segmentation (*Thong et al., 2016*), pancreas segmentation (*Cai et al., 2016*), and prostate segmentation (*Cheng et al., 2016*). Due to hardware and data constraints, recent studies use 2D CNNs (*Greenspan, Summers & Van Ginneken, 2016*). However, lymph nodes are nodular structures for which 3D information is needed to discriminate them from tubular structures, as both can appear as a blob-like structure in a 2D image. Fully 3D
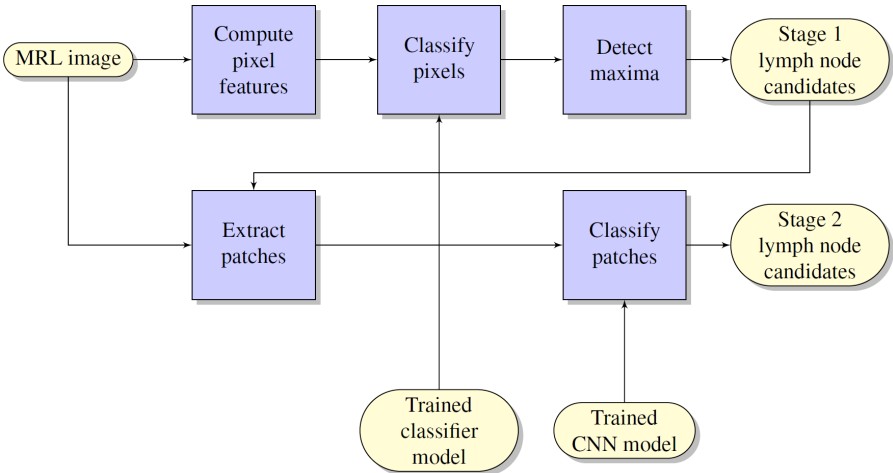

**Figure 1** Flowchart of the computerized lymph node detection method, which comprises two stages: initial lymph node detection (upper blue pipeline) and false positive reduction (lower blue pipeline).

CNNs have a strongly increased neural complexity compared to 2D CNNs, and therefore require much more training data, a disadvantage that has induced research into multi-view 2D CNNs. A three-orthogonal-views CNN ("2.5D representation") has been used for lymph node detection in CT images (*Roth et al., 2014*) and was recently expanded to random views (*Roth et al., 2016*). In pulmonary nodule classification, nine views (adding six additional oblique views to the standard orthogonal planes) were shown to outperform 3 orthogonal views in a multi-view CNN utilized for false positive reduction (*Setio et al., 2016*).

In this study, we hypothesized that a multi-view CNN can be used to reduce false positive CAD for lymph node detection in MRL images. We extended a conventional CAD system with a 1-view, 3-view, and 9-view CNN optimized for MRL imaging. We compared the performance of the extended system and the original system using bootstrapped free-response receiver operating characteristic (FROC) analysis.

## METHODS

### Lymph node detection methods

The flow chart of the lymph node detection method is shown in Fig. 1. It comprises two stages: initial lymph node detection (upper blue pipeline) and false positive reduction (lower blue pipeline).

#### *Initial lymph node detection method*

The lymph node detection stage comprised an existing prototype described in *Debats et al. (2016)*. The prototype is based on a pattern recognition system that is configured to work with MRL data sets.

A set of image features are defined, which are implemented as feature filters, each of which uses one of the available MR images as input and has a feature map as output. Several image features are computed, including

- image intensity of the VIBE, MEDIC and FLASH images, scaled by the mean $\mu$ and standard deviation $\sigma$ within each image volume
- Hessian-based blobness, vesselness, and sheetness features
- atlas features that provide positional information based on segmented pelvic anatomical structures.

After feature calculation, a voxel classification was performed which results in a lymph node likelihood between 0 and 1 for each voxel. For the voxel classification, a GentleBoost-classifier was used with regression stumps as weak learners. The number of weak learners was set to 200.

After voxel classification, a 3D likelihood map was obtained. On the likelihood map, we perform local maxima detection using a spherical window with a diameter of 10 mm, which is the maximum size of non-enlarged lymph nodes in MRL. After the local maxima detection, a merging step is performed to eliminate plateau-shaped maxima using connected component analysis. All maxima comprising more than one voxel are reduced to only the voxel nearest to their center of gravity. The location and likelihood of the remaining maxima define the lymph node detection points.

### False positive reduction method using multi-view CNNs

The false positive reduction stage is an extension to the initial lymph node detection method described in the previous paragraph. For each lymph node candidate, one or more small patches ($65 \times 65 \times 1$) were extracted around the candidate position. The set of candidate patches is presented to a convolutional neural network (CNN) classifier that discriminates between lymph nodes and false positive detections from the previous stage. We used a multi-view CNN, which takes multiple input patches across different branches. This idea was successfully explored for pulmonary nodule detection in CT (*Setio et al., 2016*) as well as for lymph node classification (*Roth et al., 2014*; *Roth et al., 2016*).

Three sets of views are considered in this paper, each one a subset of the set of views presented in Fig. 2. The 1-view set uses only the original 2D plane image information and may be limited in its ability to discriminate lymph nodes from, for example, blood vessels. The 3-view set adds two orthogonal planes which may allow incorporation of 3D structure and more surrounding information. The 9-view set provides a more finely sampled 3D view as the 3-view set and might allow better 3D shape assessment.

The CNN architecture is shown in Fig. 3 for the 3-view configuration. Each patch in the candidate set is input to a 5-layer CNN branch consisting of three convolutional layers and two max-pooling layers. The pooling operations occur after the first and second convolutional layer. Subsequently, the feature maps of the different branches were concatenated in the feature dimension and fed to the last layers of the CNN. Weights between view branches are not shared; this allows each branch to learn filters for each
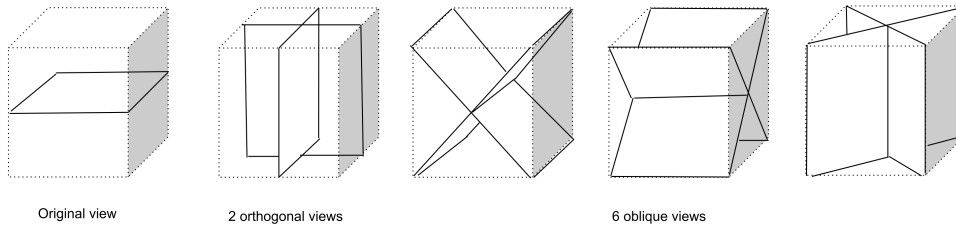

Original view        2 orthogonal views        6 oblique views

**Figure 2**  **Nine types of patch samplings used to compile a 1-view, 3-view, or 9-view set of patches that sample the surroundings of a lymph node candidate location for the CNN model.** The 1-view set comprises only the original image view, The 3-view set extends the 1-view set with 2 additional orthogonal planes, The 9-view set extends the 3-view set with six oblique planes.

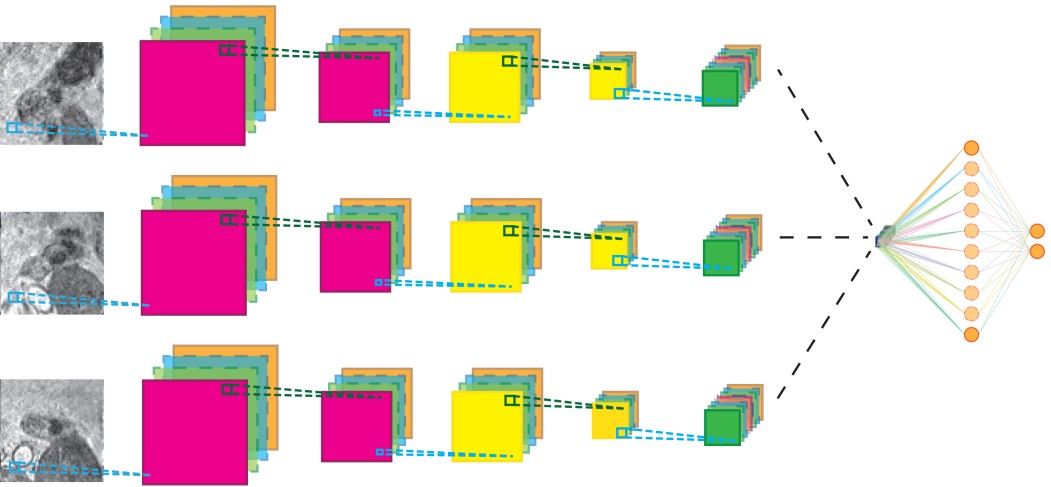

**Figure 3**  **Schematic representation of the multi-view network for three-views.** Cyan lines indicate convolution operations, dark green lines max-pooling operations. The black lines indicate a feature map concatenation. A full parameter listing is specified in Table 1.

view separately. In all layers, rectified linear units (Relu) were used as non-linearities. The detailed configuration of the CNN architecture is given in Table 1.

Network weights were optimized using standard stochastic gradient descent with a learning rate of $10^{-4}$ and a Nesterov momentum of 0.9. The loss function was categorical cross-entropy and a $\lambda_2$ of 0.0005 was used for regularization. Finally, during training, a dropout of 20% was applied to the fully connected layers. Batches of 32 multi-view patches where used during training for 200 epochs. Each epoch was defined as 200 mini-batches. Patches where randomly selected from the training set and added to a mini-batch while making sure that each mini-batch had a balanced class distribution. Training was stopped if validation accuracy did not improve for more than 20 epochs.

**Table 1 Parameters of the multi-view network.** Sizes are specified as (width, height, channels).

| Layer | Type | Filter size | Stride | Image size |
|---|---|---|---|---|
| | | Per view | | |
| 1 | Input | | | $65 \times 65 \times 1$ |
| 2 | Convolution | $5 \times 5 \times 1$ | $1 \times 1$ | $61 \times 61 \times 24$ |
| 3 | Max Pool | $2 \times 2 \times 1$ | $2 \times 2$ | $30 \times 30 \times 24$ |
| 4 | Convolution | $5 \times 5 \times 24$ | $1 \times 1$ | $26 \times 26 \times 48$ |
| 5 | Pooling | $2 \times 2 \times 1$ | $2 \times 2$ | $13 \times 13 \times 48$ |
| 6 | Convolution | $5 \times 5 \times 48$ | $1 \times 1$ | $9 \times 9 \times 96$ |
| | | After view concatenation | | |
| 7 | Fully-connected | | | $1 \times 1 \times 512$ |
| 8 | Logistic regression | | | $1 \times 1 \times 2$ |

**Table 2 Scan parameters of the MRL sequences.**

| Description | Acronym | TE (ms) | TR (ms) | Flip angle (deg) | Matrix | PS (mm) | ST (mm) |
|---|---|---|---|---|---|---|---|
| T1-weighted spin echo | VIBE | 2.45 | 4.95 | 10 | $320 \times 320$ | $0.8 \times 0.8$ | 0.8 |
| T2*-weighted gradient echo | MEDIC | 11 | 20 | 10 | $320 \times 320$ | $0.8 \times 0.8$ | 0.8 |

**Notes.**
TE, echo time; TR, repetition time; PS, pixel size; ST, slice thickness.

## Data collection
### Imaging
The lymph node detection system was evaluated using retrospectively collected MRL images of a consecutive set of patients who fulfilled the following inclusion criteria:

- Biopsy-proven prostate cancer
- Ferumoxtran-10 MRL between January 2008 and April 2010
- Successful acquisition of VIBE, MEDIC, and FLASH image volumes

A total of 240 patients were included in the analysis. Each patient underwent MR imaging enhanced with the USPIO-based lymph node specific contrast agent ferumoxtran-10 (Sinerem®, Guerbet, Paris, France) at the Radboud University Medical Center in Nijmegen, The Netherlands, as part of their clinical evaluation. The contrast agent was administered intravenously, 36 to 24 h before MR imaging was performed. All patients received a drip infusion with a duration of approximately 30 min, containing a dose of 2.6 mg Fe per kg body weight. Immediately before the MR examination, Buscopan (20 mg i.v. and 20 mg i.m.) and Glucagon (20 mg i.m.) were administered in order to suppress bowel peristalsis.

All imaging was performed using a 3.0 T MR imaging system (Magnetom TrioTim; Siemens, Erlangen, Germany). Images were acquired in the coronal plane, covering the whole pelvis. Two MR series, called VIBE and MEDIC respectively, were used in this analysis. The scan parameters are shown in Table 2. All MRL studies had sufficient image quality to be included; none needed to be discarded.

### Ethical approval and informed consent

The scientific use of clinically obtained image data was approved by the CMO Regio Arnhem-Nijmegen (the institutional review board). The IRB approval number is CMO2016-3045. All patients provided written informed consent for the use of their clinical images for research purposes.

### Annotation

To train the system and assess the performance of the automated lymph node detection, reference lymph node annotations were created for each MRL. The annotation comprised a consensus reading by two expert readers: an MD researcher specially trained in reading MRL scans (2 years of MRL experience, >300 MRLs), and an abdominal radiologist (>10 years of MRL experience, >1000 MRLs).

The T1-weighted (VIBE) sequence, which is insensitive to USPIO contrast, was used for localization and assessment of shape and size of the lymph nodes. In the VIBE images, the individual lymph nodes were interactively segmented using the application Lymph Node Task Card, developed by Siemens, Malvern, PA (USA). The T2*-weighted (MEDIC) sequence was used to assess USPIO uptake for the clinical diagnosis of the patient but was not used in our analysis. A total of 5,089 lymph nodes were annotated, 21 per patient on average. All lymph nodes visible in the images were annotated; none were excluded based on small size or otherwise.

## Training and performance evaluation

The first stage Gentleboost classifier was trained with training sets containing feature data from randomly selected equal amounts of background voxels and lymph node voxels. To avoid partial volume effects and annotation errors, edge boundary voxels were excluded from the voxel training set. This was implemented by including only training voxels that were within a four mm radius of the center of gravity of the annotated lymph node. Lymph nodes with a volume of less than 0.1 ml (approx. 195 voxels) were excluded from the training set.

The multi-view convolutional networks were trained with all lymph nodes (true positives and false negatives from the first stage) and a random subset of false positives from the first stage. For each lymph node candidate, nine image patches were extracted within a cube of $52 \times 52 \times 52$ mm ($65 \times 65 \times 65$ voxels) which encloses the candidate. The nine patches are extracted on planes corresponding to the planes of symmetry of a cube. Three of the planes run parallel to pairs of faces of the cube. These planes are commonly known as axial, coronal, and sagittal planes. The other six planes run diagonally from one edge of the cube to the opposite edge, as shown in Fig. 2.

To obtain an unbiased performance estimate, a commonly used 5-fold cross validation scheme was used. Available data were randomly split into five parts at the patient level, ensuring each part had an equal number of true positive lymph nodes. In each of the five cross-validation runs, four folds were used for training and one fold was used as the test set. In the training, three folds were used to train the CNN and one fold was used as the validation set. The predicted outputs in the 5 independent cross-validation runs were collected and used for statistical analysis.
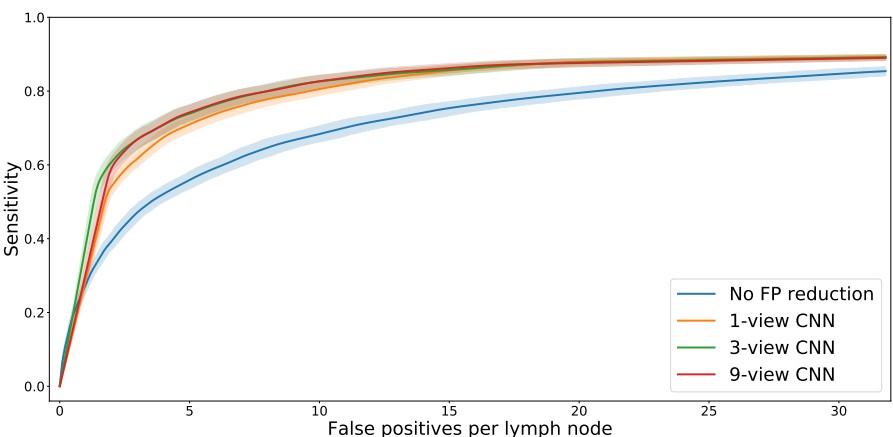

**Figure 4  FROC plot of detection performance.** The shaded areas indicate the 95% confidence intervals, obtained with bootstrapping.

## Experiments

The first experiment assessed the general effect of adding the second-stage multi-view CNN to the lymph node detection system. The diagnostic performance is determined using a free-response receiver operating characteristic (FROC) analysis. The curves are compared analytically and statistically using bootstrapping analysis.

The second experiment assessed the effect of the number of views on the performance of the lymph node detection. Multiple statistical comparisons were made between the 1-view, 3-view, and 9-view configuration.

Both experiments were performed using a one-sided bootstrap test on the partial area under the FROC curve (pAUC) ranging from 0.25 false positives per lymph node to 32 false positives per lymph node. Bonferroni correction was applied to account for multiple comparisons.

## Statistical analysis

The authors have used NumPy and SciPy to conduct the bootstrap statistical tests. A $p$-value less than 0.05 was considered statistically significant.

## RESULTS

### False positive reduction performance

The FROC analysis (Fig. 4) showed that adding the multi-view CNN increases the sensitivity of the lymph node detection at low FP levels. At a fixed level of sensitivity, the curves clearly show a reduced number of false positives. This is confirmed by the statistical analysis (Table 3). The difference in pAUC between the stage 1 system and the CNN-enhanced stage 2 lymph node detection with 1, 3, or 9 views was statistically significant ($p < 0.01$ in all cases).

The FROC analysis also showed that the differences in performance between the three multi-view CNNs are small. Statistical tests (Table 3) showed that the 3-view and 9-view systems significantly outperformed the 1-view system ($p < 0.01$ in both cases).

**Table 3  Partial area under the FROC curve (pAUC) for the four different systems.** The 95% confidence intervals are indicated between brackets. *P*-values were obtained using one-sided bootstrapped comparison of the systems after which Bonferroni correction was applied. The three rightmost columns show the *p*-values of the six pairwise comparisons. CD, candidate detector.

| System | pAUC | *p* (CD) | *p* (1-view) | *p* (3-view) |
|---|---|---|---|---|
| CD | 24.39 (23.81–24.97) | | | |
| 1-view | 27.55 (27.14–27.97) | <0.01 | | |
| 3-view | 28.02 (27.54–28.51) | <0.01 | <0.01 | |
| 9-view | 27.91 (27.47–28.34) | <0.01 | <0.01 | 0.86 |

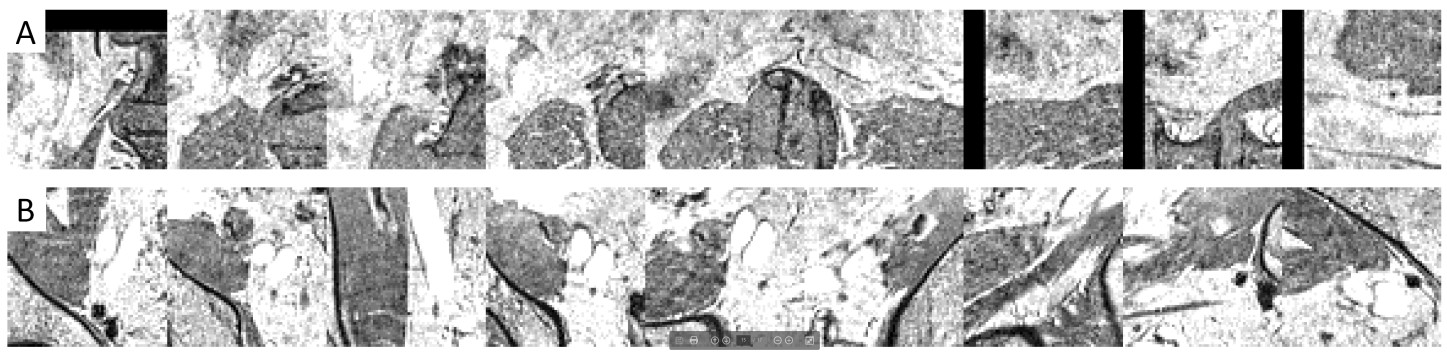

**Figure 5  Examples of false positives that were correctly dismissed by the CNN false positive reduction stage.**

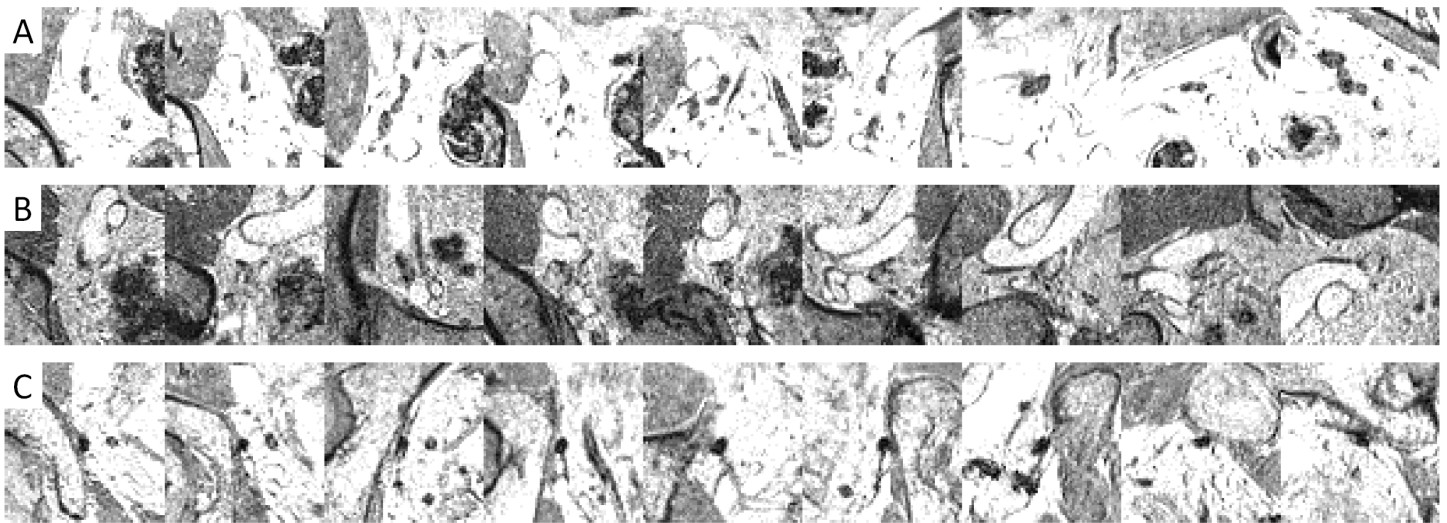

**Figure 6  Examples of image patches containing lymph nodes which were correctly retained by the CNN false positive reduction stage.**

Example images (Figs. 5 and 6) illustrated how the multi-view convolutional network correctly dismisses false positive lymph nodes and retains true positives.
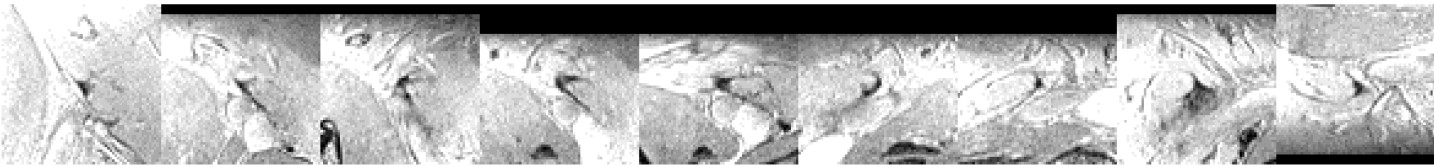

**Figure 7** Examples of image patches not annotated as lymph nodes, but with high lymph node likelihood according to the CNN.

## DISCUSSION

Magnetic resonance lymphography (MRL) can help reduce unnecessary surgery or provide image-guided treatment in the management of prostate cancer. MRL is, unfortunately, an imaging modality that is difficult to read. Tools to help assist MRL reading are essential to help make MRL part of regular diagnostic procedures. This study shows that an automatic lymph node detection tool may become more feasible when extending an existing prototype with a Deep Learning false positive reduction system. The results confirm our hypothesis and show that multi-view convolutional neural networks can improve an automated lymph node detection system for pelvic MR Lymphography images of patients with prostate cancer. The performance achieved with the new system (5-10FP/85% sensitivity) may allow clinical implementation as a tool that helps to reduce the tedious task of finding all lymph nodes.

Figure 4 shows that adding a multi-view convolutional network enables correct discrimination of a substantial number of true positive findings from false positive ones, as evidenced by the strongly improved FROC curve. The performance of the enhanced CAD system now reaches for the first time a level where we were able to observe several false positives that are likely to have been missed in the expert annotation and are in fact true positives (Fig. 7).

The increased performance is mostly due to the ability of the CNN to extract problem specific image features. The fact that most of the performance increase is already visible with a 1-view systems means that the previously built prototype lacked basic image texture features that could properly describe the aspects of the lymph node tissue and its surroundings. It is generally known that lymph nodes are usually surrounded by fatty tissue. Yet, in practice, the nodes are often observed very near vessel walls with little to no fat in between the node and the vessel wall. These kinds of heterogeneities in the surrounding anatomy make it difficult to find and implement hand-crafted features that help discriminate true lymph node regions from other regions. The CNN is able to learn image features that capture these more complex anatomical surroundings. The relatively small improvement in performance seen when adding the third dimension in a 3-view or 9-view multi-view CNN was unexpected. It shows that although the third dimension is helpful, it is the in-plane image appearance that provides the most information. This agrees with radiology practice in which an in-plane view triggers a lymph node finding and back and forth slice scrolling is used as a confirmation.

Annotation is fundamental in training machine learning systems to perform data analysis in general, and in medical diagnosis more specifically. In this study, significant effort was spent on carefully annotating the imaging data by trained experts. The advent of deep learning is starting to produce systems that operate at or above the expert level (*De Fauw et al., 2018*). This allows creating systems that can help support clinical decision making. On the other hand, the systems start to challenge the expert by detecting patterns missed by the expert during system training. In this study as well, lymph nodes were detected that the experts had missed. The consequence is that it is difficult to design a system that outperforms an expert given that the expert provided the annotations. A possible solution is to do further iterations of consensus reading whereby the system output is included in the consensus. This will likely lead to improved performance of the CNN and is part of future research.

This research confirms an earlier observation by Setio et al. that multi-view CNNs are highly suited to be used for false positive reduction of a CAD system (*Setio et al., 2016*). The performance increase in our study is similar, although at a low number of false positives the gain in performance is not as high. We believe this might be due to the fact that not all lymph nodes were annotated by the observers and as such some false positives are actually true positives. The paper by Setio et al. used the LIDC lung nodule dataset which has seen a much more elaborate consensus annotation of experts. This study did not observe a significant benefit when using 9-view CNN as compared to 3-view CNN. The difference may be attributed to the observation that for lymph nodes, it is the in-plane image appearance that provides the most information, while the CT images used for pulmonary nodule detection contain much more intersections with small vessels that are hard to distinguish from lung nodules in the in-plane image.

The amount of training data was limited. For the first stage of the CAD system, this probably didn't affect performance, but for the multi-view CNN, more training data would potentially have resulted in an increased performance of the 9-view system with respect to the 3-view system.

In conclusion, we presented a method using multi-view convolutional networks to improve a fully automated lymph node detection system for pelvic MR Lymphography. Our method has reached a level of performance where clinical implementation may help reduce reading time.

### Funding
This project was funded by Grant No. KUN2007-3971 from the Dutch Cancer Society. The funders had no role in study design, data collection and analysis, decision to publish, or preparation of the manuscript.

### Grant Disclosures
The following grant information was disclosed by the authors:
Dutch Cancer Society:  KUN2007-3971.

## Competing Interests

Henkjan J. Huisman is an Editor for PeerJ.

## Author Contributions

- Oscar A. Debats and Geert J.S. Litjens conceived and designed the experiments, performed the experiments, analyzed the data, prepared figures and/or tables, authored or reviewed drafts of the paper, approved the final draft.
- Henkjan J. Huisman conceived and designed the experiments, prepared figures and/or tables, authored or reviewed drafts of the paper, approved the final draft.

## Human Ethics

The following information was supplied relating to ethical approvals (i.e., approving body and any reference numbers):

The name of the institutional review board is CMO Regio Arnhem-Nijmegen (CMO2016-3045).

## Data Availability

The source code of the deep learning system and the four data sets are available as Supplemental Files.

## Supplemental Information

Supplemental information for this article can be found online at http://dx.doi.org/10.7717/peerj.8052#supplemental-information.

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
