# Peer review of "Lymph node detection in MR Lymphography: false positive reduction using multi-view convolutional neural networks"

_PeerJ, doi:10.7717/peerj.8052_

## Round 0.1 · original submission · Major Revisions

Dear Dr. Debats,

Your manuscript has potential to improve metastatic lymph node detection, and guide decision for therapy of patients with prostate cancer.

However, the manuscript needs to have a major revision based on the issues raised by the expert reviewers.

Please point-by-point address each reviewer's questions.

In addition, the following issues need to be addressed:

In Methods section, the sentences between lines 165-171 should be written in simple past tense instead of simple present tense.

Similarly, the Results section between lines 178-187 should be written in simple past tense.

Best Regards,

Reviewer 1 ·

Basic reporting

The article is well-written, with the few typos/language issues noted below:

* p2.l82: t1-weighted -> T1-weighted
* p4.l152: "detected om the first"
* p5.l216: "a great deal of effort" -> "significant effort"
* p5.l237: " this didn’t probably affect performanc" -> "probably didn't affect"

The literature references and related discussion could be improved. First, it is not clear what is the performance of MRL in localizing lymph nodes, and what is the clinical practice to confirm the findings. For example, is biopsy needed for confirmation? What are the implications of the incorrect localization/missed findings? Second, it is not clear whether there are any other methods for automatic detection of lymph nodes in MRL? The introduction states "A modest number of studies have been published on the development of lymph node CAD systems, most of them using CT imaging rather than MR imaging" - are there any lymph node CAD studies using MR or not? If the answer is positive, it would be ideal to see a comparison of the proposed approach with the earlier alternatives.

The authors might want to extend their review of the related literature, and, for example, consider citing the following popular article and educating the reader a bit more about the capabilities and limitations of MRL:

Hövels, AM, et al. "The diagnostic accuracy of CT and MRI in the staging of pelvic lymph nodes in patients with prostate cancer: a meta-analysis." Clinical radiology 63.4 (2008): 387-395.

The organization of the article is appropriate, figures and tables are relevant and professionally prepared.

It is not clear whether raw data shared meets the requirements of this journal. The authors share the data needed to generate the ROC plots, but not the data that was analyzed or the annotations of the image data. Journal guidelines do not appear to be specific in defining what "raw data" is in https://peerj.com/about/policies-and-procedures/#data-materials-sharing. Therefore, this aspect is deferred to the handling Editor.

Results and hypotheses stated and investigated are appropriate and self-contained.

Experimental design

The article is within the scope of the journal.

With the caveats discussed above, the research question is relevant and fills an important gap. Additional caveat to the study is the lack of clarity regarding the accuracy of the reader localization/annotation of the lymph nodes. The nodes used to train and evaluate the system were annotated by 2 readers in consensus. The performance of the system is therefore evaluated relative to the unknown reference performance of this consensus read. It would be helpful if authors discussed expected performance of the consensus read, especially since on several occasions authors note in Discussion that expert annotations could have missed some nodes. Is it possible that there are also false positives in the expert-produced annotations? This reviewer is not a clinical domain expert, perhaps those questions are obvious to the experts, but should also be clarified for more naive readers.

The description of the initial lymph node detection method (p2.l80) should be expanded. The current description is highly superficial. For example, it is not clear what it means to compute mean intensity feature of an input T1-weighted image for detection of lymph nodes. What is the meaning of the mean of all the pixels of an image? More details about the classifier are needed as well. The description should be complete for the reader to gain understanding of the approach without having to read earlier Debats et al. paper.

This reviewer does not have expertise in implementing or deploying deep learning methods. As such, it is difficult to assess the completeness of the level of detail accompanying the methods. Notably, the authors share the source code to enable replication of their results. That attachment however is lacking any documentation or user guide, and, as mentioned above, the source data used for training has not been shared. It is therefore difficult to establish whether the results could be replicated reliably, or what effort such replication endeavor would involve.

Other notes on data collection:
* inclusion criteria mention "Successful acquisition of VIBE, MEDIC, and FLASH image volumes", but there is no explanation of what this means in practice. How was "successful acquisition" defined? Were any imaging studies discarded based on image quality? If yes, how was that determination done? This point is important to evaluate the possible bias in data collection.
* no details are provided about the number of nodes annotated, number of nodes per patient, distribution of node volumes (and what percentage of the nodes were excluded due to small size)

Validity of the findings

The main question regarding the validity of this work appears to be in lacking details about the reference used to train and evaluate the system. Without knowing the performance characteristics of the readers, it is impossible to assess the absolute performance of the system, and as such - the actual possible impact of it on patient management.

Reviewer 2 ·

Basic reporting

In this study, the authors aim to determine whether multi-view convolutional neural networks (CNN) can improve a fully automated lymph node detection system for pelvic MR Lymphography (MRL) images of prostate cancer patients. They extended the fully automated computer-aided detection (CAD) system, and compare the performance of the extended system and the original system.

Experimental design

A total of 240 patient data were collected retrospectively in the Radboud University Medical Center in Nijmegen, The Netherlands from January 2008 and April 2010. This study may be interesting for clinicians who are mainly interested in prostate cancer. However, the paper needs additional works, and some minor changes are recommended before considering the paper for publication.

Firstly, they should include the information in the manuscript according to the ethical approval and consent to participate if they are available.

Validity of the findings

Secondly, I also attached the Turnitin report for plagiarism check. The similarity index at the 9th page of the Turnitin Originality Report was 25%, and this value exceeded acceptable level range between 15%-20%. Especially authors’ previous paper (Debats, O. A., Meijs, M., Litjens, G. J. S., and Huisman, H. J. (2016). Automated multistructure atlas-assisted detection of lymph nodes using pelvic mr lymphography in prostate cancer patients. Medical Physics, 43(6):3132.) which was cited in the manuscript has the highest similarity with 13%.

Thirdly, the details of the methods are not available in the manuscript. In addition to this, it is not obvious why they prefer to use CNN method rather than other machine learning methods; or what is the reason to use 5-fold cross validation instead of 10-fold cross validation.

Lastly, authors should include the name of the statistical software used in the statistical analysis.

Annotated reviews are not available for download in order to protect the identity of reviewers who chose to remain anonymous.

---

## Round 0.2 · accepted · Accept

Dear Dr. Debats,

I am glad to inform you that your manuscript has obtained favorable comments from the expert reviewers.

It is now acceptable for publication.

However, the changes indicated in the annotated PDF file need to be performed.

Best Regards,

Reviewer 1 ·

Basic reporting

The language use is clear and professional.

Compared to the initial submission, the authors improved the rigor of reporting for the related work analysis, and provide significantly more details about the methodology, as suggested by the reviewers.

Experimental design

The experimental design follows the conventional approaches accepted in the community, and the experiments are well-justified. The manuscript is accompanied by the source code for the key components of the developed tools, which should allow the reader (who should be an expert in this field) to re-implement the methodology.

Validity of the findings

The original imaging datasets are not shared, which is (unfortunately) common in radiology, and in this situation is justifiable due to the lack of provisions allowing to share the dataset at the authors' institutions. I have nothing else to add in addition to the original review of this aspect.

Additional comments

The authors revised the paper appropriately to address the concerns expressed for the original submission.

Reviewer 2 ·

Basic reporting

no comment

Experimental design

no comment

Validity of the findings

no comment

Additional comments

My questions and concerns were addressed.